# Analysis of Factors Related to Mental Health, Suppression of Emotions, and Personality Influencing Coping with Stress among Nurses

**DOI:** 10.3390/ijerph19169777

**Published:** 2022-08-09

**Authors:** Anna Maria Cybulska, Kamila Rachubińska, Marzanna Stanisławska, Szymon Grochans, Aneta Cymbaluk-Płoska, Elżbieta Grochans

**Affiliations:** 1Department of Nursing, Pomeranian Medical University in Szczecin, 71-210 Szczecin, Poland; 2Department of Biochemistry and Medical Chemistry, Pomeranian Medical University in Szczecin, Powstańców Wielkopolskich 72 Street, 70-111 Szczecin, Poland; 3Department of Gynecological Surgery and Gynecological Oncology of Adults and Adolescents, Pomeranian Medical University, 70-111 Szczecin, Poland

**Keywords:** coping strategies, health, nursing, stress

## Abstract

(1) The specificity of a nurse’s work, apart from performing medical procedures, is characterized by intensive contacts with other people. Stress is an inevitable part of a nurse’s job and can affect their physical and mental health. Thus, strategies for coping with stress play an important role in improving health or well-being by reducing the level of stress. (2) The aim of the study was to evaluate the impact of coping strategies in predicting the overall health of nurses. The study also assessed the impact of personality traits and emotional control (anger, depression, anxiety) on the choice of coping with stress among nurses. (3) The study included 811 nurses from the West Pomeranian Voivodeship, with an average age of 40 (SD = 9.8), working mainly in hospitals (82%). The research was carried out with a diagnostic survey method, using the Coping Inventory for Stressful Situations questionnaire, NEO-Five Factor Inventor, Courtland Emotional Control Care Scale, General Health Questionnaire 30, and a demographic questionnaire. (4) Among the surveyed nurses, the dominant style of coping with stress was the style focused on emotions (MT 0.43), followed by the style focused on avoidance (MT 0.42). There were mental problems among 46.1% of the respondents. Nurses with mental problems according to GHQ-30 were characterized by a high intensity of coping styles focused on emotions (30.2%), avoiding (18.7%), and engaging in alternative activities (32.3%) (*p* = 0.000). (5) Most of the surveyed nurses have a tendency to cope with stress through an emotional-focused style, which may be associated with a higher level of occupational stress.

## 1. Introduction

Nurses play a very important role in healthcare, unfortunately, the difficulties related to their work expose them to various health hazards. Work-related stress, combined with everyday stress, has a negative impact on many aspects of nurses’ lives. Excessive physical and mental strain on nurses contributes to many health complications. Work-related stress is one of the most serious health and safety challenges. It is ubiquitous and often cannot be prevented, but it can also act as a motivating factor. Research shows that every fourth employee in the general labor environment experiences stress. It is associated with many costs, not only economic, but also in the form of health disorders. Stress in the workplace can affect anyone, regardless of the position held, the sector of work or the size of the institution in which they work [1,2].

Nurses are the most numerous group of healthcare professionals in almost all countries in the world. The working conditions of the nursing staff impose high mental and physical demands on this professional group. Nurses, while performing their duties, must show many professional and personal qualities, and above all, the ability to deal with stress. Assessment of the impact of stress on the work and health of nurses can be an important element in preventing many threats, including burnout, which refers to the emotional depletion and loss of motivation that result from prolonged exposure to chronic emotional and interpersonal stressors on the job. Nurse burnout is the state of mental, physical, and emotional exhaustion caused by sustained work-related stressors such as long hours, the pressure of quick decision-making, and the strain of caring for patients who may have poor outcomes [3].

Stress is a potential determinant of health because it determines the mental state of a person, which is a reflection of the reaction to environmental conditions. A special place full of stressors is the work environment [4]. Interactions between stress and employees’ health problems are becoming important. Each person experiences stress differently, which is an indispensable element of work and personal life. The threshold of resistance to the impact of various stressors varies from individual to individual. The pace of the modern world is a factor contributing to the occurrence of situations that cause psychological discomfort [5].

The group particularly exposed to stressors in the workplace are nurses who engage in social, psychological or physical problems of their patients, often working in strenuous conditions and under the influence of strong emotions. The literature review of the subject shows that the most effective style of dealing with professional problems among nurses is the style focused on the task and seeking social contacts. Due to their profession, the ability to control emotions is undoubtedly required among nurses because the health, life, and safety of others depend on how effectively they cope with stressful situations [6]. Nurses who tend to deal with stress through an emotion-focused style are more likely to report higher levels of occupational stress. Maslach states that nurses and midwives belong to the professions exposed to the burnout syndrome, which occurs in a situation of chronic stress typical of social professions [7]. The consequence of long-term stress exposure is the risk of psychosomatic disorders. The intensification of this phenomenon was noted along with the COVID-19 pandemic, when a significant impact of stress on the mental health of nursing staff was observed [8,9,10,11,12,13].

The World Health Organization (WHO) defines work-related stress as the response that people may have due to emerging work-related needs and pressures that do not match their knowledge or skills, and that pose a significant challenge to their ability to cope with them [14]. Managing stress is defined as cognitive behavior and problem solving that are used to tolerate, minimize or eliminate stress. The use of appropriate stress coping strategies allows to reduce the risk of certain stress-related diseases [15].

Many studies confirm that personality traits are an important factor influencing the identification or response to stressful events [16]. One of the most widespread theories of personality is the five-factor personality model known as the Big Five. This model covers the five main dimensions of personality: neuroticism, extraversion, openness to experience, agreeableness, and conscientiousness. Neuroticism is characterized by a susceptibility to experiencing negative emotions, such as anxiety, anger, insecurity, impulsiveness, and sensitivity, especially to psychological stress. Extroverted people are self-confident, friendly, talkative, playful, stimulate-seeking and are characterized by enthusiasm, energy, and a cheerful mood. Openness to experience is a positive evaluation of life experiences, tolerance towards novelty, cognitive curiosity, impulsiveness, originality and creativity. Agreeableness is characterized by cooperation, morality, sympathy, low self-confidence, a high level of trust in others, a tendency to be happy. It is worth noting that personality traits allow you to react to different situations in a specific way; therefore, maladaptive personality traits, e.g., neuroticism, are associated with increased exposure to stressful life events and thus may make people more susceptible to experiencing negative emotions or frustration [17].

Whereas adaptive personality traits (e.g., conscientiousness) facilitate coping with stresses of everyday life [18].

Managing stress is a regulatory process that can reduce the negative feelings re-sulting from stressful events. Moreover, it is a dynamic process that changes over time in response to changing requirements or the assessment of the situation [19]. There are three main styles of coping with stress in the literature: task-focused, emotion-focused, and avoidance-focused. Task-focused stress management is the basic adaptive control style that allows to control emotions and focus on the implementation of a specific task. The style focused on emotions is associated with experiencing and trying to release emotions related to a stressful situation. Avoidance-focused coping is a cognitive and behavioral effort aimed at minimizing, denying or ignoring stressful situations [16,20].

Personality traits have important implications for the intrapsychic and interpersonal resources of an individual, and therefore, they fall within the options of coping with stressful situations. The literature review of the subject shows that adaptive personality traits are significantly positively associated with active coping styles. While maladaptive personality traits are positively associated with avoidance-focused coping with stress. Based on this information, it can be observed that people with a maladaptive personality are more likely to experience psychological distress because they probably use a maladaptive coping style, such as coping with avoidance [21]. Therefore, the most effective style of dealing with professional problems among nurses is a style focused on the task and seeking social contacts. Due to their profession, the ability to control emotions is undoubtedly required among nurses because health, life, and safety of others depend on how effectively they deal with stressful situations [22]. Nurses who tend to deal with stress through an emotion-focused style are more likely to report higher levels of occupational stress.

The aim of the study was to evaluate the impact of coping strategies in predicting the overall health of nurses. The study also assessed the impact of personality traits and emotional control (anger, depression, anxiety) on the choice of style of coping with stress among nurses.

## 2. Materials and Methods

### 2.1. Setting and Design

The study was conducted among 811 nurses who agreed and met the inclusion criteria, which included:Employment in one of the hospitals in the West Pomeranian Voivodeship;Bachelor’s or master’s degree in nursing;At least one year of work experience.

The selection of the group was random and resulted from the particular exposure to stress of the respondents resulting from the specificity of the profession, i.e., contact with sick, suffering and dying people, heavy workload, and deficiencies in therapeutic teams.

The research was carried out using the traditional method, by disseminating paper versions of questionnaires among nurses working in the West Pomeranian Voivodeship.

The study was conducted in accordance with the guidelines of the Helsinki Declaration and approved by the Ethics Committee of the Pomeranian Medical University in Szczecin (KB-0012/102/12/2013). Our study was conducted with ethical considerations in mind. Informed consent was required and participation in the study was voluntary. In addition, participants were guaranteed anonymity and confidentiality, and the possibility of withdrawing from the study at any stage.

### 2.2. Research Instrument

The study used a diagnostic survey method with the use of the survey technique. The following questionnaires were employed to collect the research material:**Coping Inventory for Stressful Situations (CISS)** is a tool for assessing how to cope with stress and it consists of 48 statements about the different behaviors people undertake in stressful situations. The respondent determines the frequency with which he/she takes a given action in difficult and stressful situations on a 5-point scale. The results are presented in three scales, where the style focuses on: task, emotions, and avoidance. The latter takes two forms: engaging in alternative activities and seeking social contacts. The range of results for task, emotions, and avoidance from 16 to 80 points, for engaging in alternative activities the range is 8–40, and for seeking social contacts 5–25. The results are verified using sten standards for 3 age groups (16–24, 25–54, 55–79). The reliability of the Polish version of the CISS questionnaire, measured with the Cronbach’s alpha coefficient, ranges from 0.72 to 0.92 [23].**The NEO-FIVE Factor Inventory (NEO-FFI)** is a tool for the diagnosis of personality traits taking into account the Big Five model, i.e., the five-factor personality model created by Costa P.T and McCrae R.R. The items in the questionnaire are 60 self-report statements, the truthfulness of which is assessed by the respondent on a five-point scale. These items create 5 measuring scales: neuroticism, extraversion, openness to experience, agreeableness, and conscientiousness. The range of results in each of the subscales is 0–48 points. Depending on the number of stens showing the intensity of each trait, the respondent obtains a high (7–10), average (4–6) or low (1–3) result for each of the 5 scales. The reliability of the Polish version of the NEO-FFI questionnaire, measured with the Cronbach’s alpha coefficient, ranges from 0.68 to 0.82 [24].**General Health Questionnaire (GHQ-30)** is a self-assessment instrument used for the preselection of people with non-psychotic mental disorders and as a tool for measuring nonspecific mental distress. The questionnaire includes questions that the respondents independently refer to their own situation and mental state. In the opinion of D. Goldberg, the sum of the points obtained as a result of using the questionnaire should be treated as a quantitative measure of the severity of non-psychotic disorders of mental functions, referring to the theoretical concept of the axis running from the pole of complete mental health to the pole of severe mental disorders. The greater the number of GHQ points, the greater the likelihood of psychiatric disorders. The GHQ is a self-esteem scale, so the severity of anxiety and depression as well as the assessment of functioning are subjective. The way of experiencing an individual sensitivity significantly affects the sum of GHQ points. People with a greater sense of illness score higher sums of points. The research used the GHQ score method, thanks to which it is possible to assess the general state of health. Health disorders occur when the respondent obtains more than 4 points. The reliability of the Polish version of the NEO-FFI questionnaire measured with the Cronbach’s alpha coefficient was 0.97 [25].**CECS (Courtland Emotional Control Care Scale)** is a tool that measures the subjective control of anger, anxiety, and depression in difficult situations. The respondent determines the frequency of occurrence of the given manner of expressing his emotions on a four-point scale from “almost never” (1 point) to “almost always” (4 points). The results are calculated separately for each of the three subscales. The range of results is 7–28 points. After summing up all three subscales, we obtain a general indicator of emotional control, which is within the range of 21–84 points. The higher the score, the greater the degree of negative emotion suppression. Cronbach’s alpha internal consistency index was 0.80 for anger control, 0.77 for depression, 0.78 for anxiety, and 0.87 for the cumulative emotional control index [26].Self-written questionnaire, containing questions about sociodemographic variables and variables related to the work of a nurse.

### 2.3. Statistical Analysis

The statistical analysis was performed with the use of the Excel 2007 (Microsoft Corporation, Albuquerque, NM, USA) spreadsheet and Statistica version 7 (TIBCO Software, Palo Alto, CA, USA). The collected data were analyzed using the r-Pearson correlation coefficient. The *p* value < 0.05 was adopted as statistically significant.

Descriptive statistics (count, mean, medians, modes, fashion count, maximum, minimum, lower and upper quartile, range, quartile range, standard deviations, proportions) and statistical inference were used to calculate and interpret data, and relationships between variables [27]:χ^2^ statistic (chi-square)—used to investigate the relationship between two qualitative or quasi-quantitative features. The null hypothesis assumed the independence of the analyzed features, while the alternative hypothesis stated the existence of a statistically significant relationship between them. The results of the analysis were a multi-way table with the chi-square statistic and the *p*-value calculated for it. The null hypothesis was rejected in favor of the alternative if the statistical significance p was greater than the assumed significance level α (α = 0.05). When the test probability exceeded α, it was found that there were no grounds for rejecting the null hypothesis;*p*-value—used to obtain the values of the selected statistics, which were observed assuming that the null hypothesis is satisfied;Sample randomness test—Stevens series test—aimed at checking whether the recorded results in this respect can be generalized to a larger number of cases. Before applying the procedures of statistical inference, it allows to make sure that the collected observations meet the postulate of randomness of the sample.

The collected data on personality traits according to the NEO-FFI Questionnaire and the styles of coping with stress according to the CISS Questionnaire were subjected to a multivariate analysis based on linear ordering according to A. Balicki’s pattern, which allows to organize the set of features of multidimensional objects. The pattern method makes it possible to create a hypothetical pattern with respect to which the distances from the real points are determined. The model becomes a certain ideal, but unreal model object with the best values of the variables: for positive variables—the highest, and for negative variables—the lowest. The smaller the distance between the object and the pattern is found, the more similar the object and the higher the level of complex phenomena of a given object. The taxonomic measure is the synthetic quantity calculated on the basis of the distance from the standard, which is the resultant of all variables characterizing the tested objects.

## 3. Results

### 3.1. Characteristics of the Respondents

The research sample consisted of 811 nurses who correctly completed the questionnaires. The mean age was 40 years (SD = 9.8). The vast majority of the respondents were: people with a formal relationship (63.5%), people with higher education, undergraduate (45.5%), and people living in a city with more than 100,000 inhabitants (47.1%) (Table 1).

### 3.2. Assessment of Styles of Coping with Stress, Mental Health, Personality Traits, and the Intensity of Emotions among the Respondents

Due to the fact that many work-related stress factors were perceived by nurses, the methods of coping with stress with the use of the CISS questionnaire were assessed. The highest mean was obtained in the style focused on emotions (M = 5.6) and engaging in substitute activities (M = 5.6). Based on the multivariate analysis, it was shown that the dominant style of coping with stress was the one focused on emotions (MT 0.43), followed by the style focused on avoidance (MT 0.42) (Table 2).

Personality traits of nursing staff were assessed in the study using the NEO-FFI questionnaire. The highest mean was obtained for conscientiousness (M = 6.4), while the multivariate analysis showed that the dominant personality among nurses was extroversion (MT = 0.5) (Table 2).

The study assessed the mental state of the surveyed nurses on the basis of the data obtained from the General Health Questionnaire GHQ-30, which showed that 53.9% of the nurses had no mental health disorders (Table 3).

The research analyzed emotional control with the use of the CECS Emotion Control Scale. Among the respondents, suppression of emotions was at an average level, of which anxiety was the dominant emotion felt. Both anger and depression were suppressed to a similar level (Table 3).

### 3.3. The Relationship between the Mental State According to the GHQ-30 Questionnaire and the Styles of Coping with Stress According to the CISS Questionnaire

The study analyzed the relationship between mental health according to the GHQ-30 Questionnaire and the styles of coping with stress according to the CISS Questionnaire. Statistically significant correlations between the mental health of nurses and the style focused on emotions, the style focused on avoidance, and engaging in substitute activities have been demonstrated.

Studies have shown that 9.15% of people in positive health had a low score in an emotional-focused style. In the case of healthy respondents, low results in the avoidance style were achieved by 8.9%, and in the case of high results in engaging in alternative activities, by 8.3% of the respondents. Moreover, people characterized by mental health disorders according to the GHQ-30 questionnaire were characterized by a high intensity of the style focused on emotions—30.2% (*p* = 0.000), the style focused on avoidance—18.7% (*p* = 0.002), and on engaging in alternative activities—32.3% (*p* = 0.000). No statistically significant relationships were found between the general health status according to the GHQ-30 questionnaire and the style focused on the task and the search for social contacts (Table 4).

### 3.4. The Relationship between Personality Traits According to NEO-FFI and Styles of Coping with Stress According to the CISS Questionnaire

The study analyzed the relationship between personality traits (extraversion, openness, conscientiousness, agreeableness, neuroticism) according to the NEO-FFI Questionnaire and the styles of coping (task-oriented coping, emotion-oriented coping, avoidance-oriented coping, seeking social contact, engaging in alternative activities) with stress according to the CISS Questionnaire.

On the basis of the collected data, statistically significant relationships (*p* > 0.05) were found between the choice of the style focused on the task and conscientiousness and openness to experience. It was observed that 44.84% of people with a low level of the task-focused style were characterized by a low level of conscientiousness, while 28.83% were characterized by a low level of openness to experiences. In the studies, no statistically significant relationships were found between task-focused style and neuroticism, extraversion, and agreeableness (Table 5).

Data analysis showed statistically significant relationships between the choice of the style focused on emotions and neuroticism, extraversion, agreeableness, conscientiousness, and openness to experiences. It was observed that 26.15% of the respondents who achieved high intensity in the emotional style had high levels of neuroticism. It was observed that 24.32% of people with a high level of the style focused on emotions were characterized by a low level of extraversion, and 22.81% of people with a low level of openness to experiences. Moreover, among people with low intensity of the style focused on emotions, 32.43% were characterized by low results in agreeableness, and in the case of 37.93% of respondents—with low results in conscientiousness (Table 5).

Our research showed statistically significant relationships between the choice of avoidance-focused style and neuroticism, extraversion, agreeableness, and conscientiousness. Avoidant-style highs were high on 16.04% in neuroticism and high on 17.24% in extraversion. People with a high level of avoidance style were characterized by low intensity—22.52% in the case of agreeableness and 27.59% in the case of conscientiousness. The collected data did not show any statistically significant correlation between the style focused on avoidance and openness to experience (Table 5).

Based on the collected data, statistically significant correlations were found between the choice of engaging in alternative activities, and neuroticism, extraversion, agreeableness, conscientiousness, and openness to new experiences. Among those who achieved high scores in engaging in alternative activities, 33.16% had a high score on neuroticism. It was observed that 25.73% of nurses with a high level of style focused on engaging in substitution activities had low levels of extraversion. Among the subjects with a high score in engaging in alternative activities, 34.23% were characterized by low scores in agreeableness, 50% had little degree of conscientiousness, and 24.69% had a high level of openness to experience (Table 6).

The research results showed a statistically significant relationship between the search for social contacts and agreeableness and conscientiousness. Among the respondents, 23.91% of people with high intensity in seeking social contacts achieved high results in agreeableness, and 21.40% achieved high results in conscientiousness. No statistically significant relationships were observed between seeking social contacts and neuroticism, extraversion and openness to new experiences (Table 6).

### 3.5. The Relationship between the Level of Suppression of Emotions According to the CECS Emotion Control Scale and the Styles of Coping with Stress According to the CISS Questionnaire

The study analyzed the relationship between the level of emotional suppression according to the CECS Emotion Control Scale and the styles of coping with stress according to the CISS Questionnaire.

Statistically significant relationships were observed between the level of anxiety suppression and the style focused on emotions. In the case of 28.26% of respondents who scored low in the emotional style, the level of anxiety was high. The research did not show any statistically significant relationships between suppression of anxiety and other styles of coping with stress according to the CISS Questionnaire (Table 7) and between suppressing depression and anger and styles of coping with stress (*p* > 0.05).

## 4. Discussion

### 4.1. Assessment of the Styles of Coping with Stress of the Respondents

The results of our own research indicate that the dominant style of coping with stress among the surveyed nurses is the one focused on emotions, followed by the style focused on avoidance. Research by Nyklewicz W. et al. [28] and Perek et al. [29] showed that the dominant style of coping with stress among nurses is the task response to stress. Nurses used constructive ways of coping with stress, which consisted of making efforts to solve the problem through cognitive transformations or attempts to change the situation and the implementation of tasks entrusted to them. These reports are consistent with the research results by Wilczek-Rużyczka and Król [30], which emphasize that nurses who tend to cope with stress through the style focused on emotions more often declare a higher level of occupational stress. Research by Isa et al. [31] showed that problem-oriented coping strategies were the most common ones used by nurses. Similar results were also obtained by Fiske et al. [32]. Probably the choice of this stress coping strategy results from a sense of control through careful step-by-step planning, which allows for effective management of stress factors. Moreover, in the study by Chang et al. [33], it was observed that people using problem-oriented coping strategies showed a higher index of mental health, which means lower risk of mental health disorder.

The studies by Healy et al. [34] showed a significant positive relationship between stress in nurses and mood disorders, and a significant negative relationship between stress and job satisfaction. It was observed that the use of avoidance and the perception of work overload is a significant predictor of mood disorders. In addition, studies by Deklava et al. [35] showed that one of the dominant coping strategies among nurses is the emotional coping strategy. It is worth noting that nurses who used emotion-oriented coping strategies showed higher psychological competences and significantly better professional behavior and personality traits. Emotion-oriented coping strategies are preferred by people whose personality allows them to easily enter and maintain a state of emotional arousal in response to or anticipation of emotionally charged stressful events [28]. In addition, the meta-analysis by Penley et al. [36] showed that people using emotional coping strategies have worse health outcomes because they choose negative styles of coping with emotions, such as alcohol and drug use, and the root cause of stress is not removed.

In the case of studies by Wang et al. [37], it was shown that workload, lack of support, and inadequate preparation are the most common stressors for surgical nurses, who most often chose coping strategies that can be characterized as avoidant, confrontational, and optimistic. All of these strategies are considered to be the most effective stress reduction strategies.

Research by Betke et al. [38] conducted among Polish nurses during the COVID-19 pandemic showed that the respondents most often declared using strategies which included the active ways of coping with stress, focused on the problem.

In the case of the studies by Kupcewicz et al. [39], it was observed that nurses with increased positive orientation more often used both problem-focused strategies (such as planning and active coping) and strategies reducing stress and negative emotions (such as positive revalidation, seeking support, and emotional acceptance).

On the other hand, studies by Portero de la Cruz [40] showed that the most frequently used coping strategy was problem-focused coping, and the least frequent coping by avoidance, which is consistent with the results of other studies [41,42,43,44].

### 4.2. Assessment of the Mental Health of the Respondents

The analysis of authors’ own material shows that 53.9% of the respondents do not have a risk of mental health disorders. In the case of the analysis of the results of measuring health condition (GHQ-12), research by Nyklewicz et al. [20] showed a positive dimension of mental health in over half of the surveyed nurses (61%). In turn, Makowska and Merecz noted that mental health disorders occurred in 26.6% of the respondents [25]. Studies by Bazazan et al. [45] showed that respondents achieved a GHQ-12 score above the cut-off point for about one-third of the inpatient nurses surveyed. This means that mental distress is fairly common among this group of health professionals and broadly agrees with findings in other countries. Both studies by Su et al. [46] and Suzuki et al. [47] showed that there is a high prevalence of psychiatric disorders among nurses working in a hospital. Research by Chatziioannidis et al. [48] carried out on health care workers who had been harassed or witnessed intimidation of others found that these people had higher GHQ-12 scores, indicating psychological stress. Similar results were obtained by Ramirez et al. [49]. The problem of mental health impairment among health care workers is large, as demonstrated by numerous studies. A study by Portero et al. [50] conducted among 235 doctors and nurses showed that 32.3% of professionals had mental disorders. Even clearer differences were observed in the studies by Sanchez-Lopez et al. [51] who showed that nursing staff achieved significantly worse results in terms of mental health than the entire population.

Research by Javadi-Pashaki [52] showed that half of the respondents had health disorders. In addition, studies by Noorian et al. [53], Sahraiana et al. [54], and Maghsoudi et al. [55] confirm the occurrence of mental health problems among nurses. During the COVID-19 pandemic, an increase in mental health problems was particularly noted among nursing staff [7,8,9,10,11,12].

On the other hand, research by Betke et al. [56] showed that among practicing nurses, there are three specific types of health conditions of nurses, i.e., “healthy individuals”, nurses “with disorders”, and “malcontents with health problems”. Healthy people are nurses with the best somatic health and the lowest burden of chronic diseases. The “disturbed” nurses represented poor somatic health but the best mental health. Moreover, these nurses enjoyed physical well-being. “Malcontents with health problems” are the group of nurses where the most risk factors for physical and mental well-being have been identified. These nurses reported the highest burden of chronic diseases and a stronger tendency to report somatic complaints. Moreover, nurses from this group opted for the least adaptive strategies of coping with stress.

### 4.3. Assessment of the Suppression of Emotions among the Respondents

The results of our own research indicate that the surveyed nurses are characterized by an average level of suppression of emotions. Research of Nyklewicz et al. [28] showed that 74% of respondents experienced above-average severe anxiety. Most nurses used anger, depression, and anxiety suppression. Most of the respondents suppress their emotions to an average extent. Anger is suppressed moderately and heavily by 69% of people, depression by 72%, and anxiety by 76%.

In recent years, more and more attention has been paid to the psychological adaptation of health care workers. Their work is associated with emotions that significantly affect the daily functioning and life of the medical staff. In addition, it has been reported that nurses often find it difficult to regulate their emotions because of emotional dissonance, which is the difference between the emotions they experience and those they need to express. Research by Han et al. [57] showed that among nurses who reported significant stress, 44.4% chose anger as the most frequently experienced emotion and suppressed anger more than expressed it. Additionally, Cox et al. [58] showed that women who cannot properly express their anger and use indirect methods of coping with anger or suppress their anger are more prone to depression and anxiety than women who express this anger. The suppression of negative emotions is associated with ruminations, which may eventually cause, maintain or worsen depressive and anxiety symptoms [59]. Research by Kim et al. [60] confirms that the work of a nurse influences the intensification of depressive and anxiety symptoms in the mechanism related to the suppression of anger.

### 4.4. The Relationship between the Styles of Coping with Stress According to the CISS Questionnaire and Selected Variables (General Health, Personality, Suppression of Emotions)

Our own research has shown statistically significant relationships between the styles of coping with stress according to CISS and selected variables such as general health, personality or suppression of emotions (anger, anxiety, depression).

It was observed that people with mental health disorders according to the GHQ-30 questionnaire were characterized by a high intensity of the style focused on emotions—30.2% (*p* = 0.000), the style focused on avoidance—18.7% (*p* = 0.002), and on engaging in alternative activities—32.3% (*p* = 0.000).

Pouranghash Tehrani et al. [61] showed that task-oriented and emotional coping strategies have the largest share in predicting mental health. In turn, Livarjani et al. [62] noted that, among coping strategies, the emotional strategy was a significant predictor of health, and it was found that with the increase in the emotional strategy, the general health condition deteriorated. In the case of the studies by Iannello and Balzarotti [63], it was observed that task-focused or avoidance-focused styles were predictors of lower work-related stress, while the emotional-focused style was associated with higher levels of stress.

The authors’ own research showed that 44.84% of people with a low level of the task-focused style were characterized by a low level of conscientiousness, while 28.83% were characterized by a low level of openness to experiences. The respondents who achieved high intensity in the emotional style, in this case, 26.15% of the respondents, were characterized by a high level of neuroticism, while 24.32% had a low level of extraversion, and 22.81% had a low level of openness to experience. Persons reaching high-intensity style focused on avoidance tended to be high in the case of 16.04% in neuroticism and high in the case of 17.24% in extraversion. People with a high level of avoidance style were characterized by low intensity—22.52% in the case of agreeableness and 27.59% in the case of conscientiousness. Moreover, in the case of high achievers in engaging in alternative activities, 33.16% were characterized by a high score in neuroticism, and 25.73% were characterized by a low level of extraversion. Among the respondents with a high score in engaging in alternative activities, 34.23% were low in agreeableness, 50% had little conscientiousness, and 24.69% had a high level of openness to experience. In addition, 23.91% of respondents with high levels of socializing scored high on agreeableness and 21.40% scored high on conscientiousness.

Studies by Portero de la Cruz et al. [41] showed that problem-focused coping was negatively correlated with depression and social dysfunction. It should also be noted that a significant positive correlation was found between emotional-focused counseling and depression. The use of adaptive coping styles has a positive impact on physical and mental well-being, coping with stress, and overall performance among health care workers, which is associated with improved quality of care, greater patient safety, and a decrease in health care costs [43,64].

Nurses, due to the specific working conditions, struggle with many physical and mental problems. Prolonged and continuous stress is detrimental to the health of nurses and contributes to organizational inefficiencies, so it is important to introduce some intervention measures that enable nurses to cope with stress. It is important to create a friendly, supportive, and collaborative working environment.

## 5. Limitations

First, the study used a self-report questionnaire to which the respondents could answer in a socially desirable way. The research assumption was, however, based on the respondents’ trust and their understanding of the questions. Secondly, the selection of respondents was random. Moreover, we consider a small group of surveyed nurses to be a limitation of the conducted study. In the future, it would be worth re-examining a larger group to obtain greater generalization of results. We also see that it would be interesting to compare nurses to other health professionals such as doctors and paramedics in terms of their coping strategies or overall health.

## 6. Conclusions

Mental health problems were almost entirely translated into the use of passive coping with stress strategies by the studied nurses (based on emotions, avoidance, and engaging in substitute activities), while more than half of the respondents who did not have problems related to mental disorders were characterized by the use of active strategies. The use of active strategies can also be noticed in people characterized by conscientiousness and openness to experience, while passive strategies were characterized by the nurses studied with severe neuroticism. Similarly, in the case of emotional control, this suppression of anxiety was characteristic of the respondents using the emotional-based style of coping with stress. It is important for nurses to learn the right ways of coping with stress and use them correctly depending on the situation related to working with the patient. Proper application of stress coping strategies by nurses will improve the safety and quality of nurses’ work, and thus, the safety of patients and the quality of caregiving.

## Figures and Tables

**Table 1 ijerph-19-09777-t001:** Sociodemographic variables.

Variables (*N* = 811)		*n*	%
Age	≤29 years old	171	21.1
	30–39 years old	167	20.6
	40–49 years old	333	41.1
	≥50 years old	141	26.3
Marital status	Formal relationship	515	63.5
	Single/divorced/widowed	296	36.5
Education	Medical high school	157	19.4
	Medical studies	75	9.2
	Higher education (Bachelor)	369	45.5
	Higher education (Master)	210	25.9
Place of residence	village	155	19.1
	city with up to 10,000 people	62	7.6
	city with 10,000–100,000 people	212	26.1
	city with over 100,000 people	382	47.1
Work experience [years]	≤14 years	274	33.8
	15–24 years	288	35.5
	25–34 years	220	27.1
	≥35 years	29	3.6

*N*, whole cohort size; *n*, number of; %, percent.

**Table 2 ijerph-19-09777-t002:** Analysis of the styles of coping with stress according to CISS and personality traits according to NEO-FFI.

Variables (*N* = 811)		M ± SD	Me	Min.–Max.	Q1–Q3	R	MT
CISS (stens)	Emotion-oriented coping	5.6 ± 1.3	6.0	2.0–10.0	5.0–6.0	8.0	0.43
	Avoidance-oriented coping	5.4 ± 1.4	5.0	1.0–10.0	5.0–6.0	9.0	0.42
	Engaging in alternative activities	5.6 ± 1.5	5.0	1.0–10.0	5.0–7.0	9.0	0.37
	Task-oriented coping	4.7 ± 1.5	5.0	1.0–10.0	4.0–6.0	9.0	0.35
	Seeking social contact	5.1 ± 1.5	5.0	1.0–10.0	4.0–6.0	9.0	0.35
NEO-FFI (stens)	Extraversion	5.8 ± 1.8	6.0	1.0–10.0	5.0–7.0	9.0	0.50
	Openness	5.5 ± 1.8	5.0	1.0–10.0	4.0–7.0	9.0	0.44
	Conscientiousness	6.4 ± 2.1	6.0	1.0–10.0	5.0–8.0	9.0	0.40
	Agreeableness	5.8 ± 2.2	6.0	1.0–10.0	4.0–7.0	9.0	0.37
	Neuroticism	4.7 ± 2.2	5.0	1.0–10.0	3.0–6.0	9.0	0.34

M—mean; SD—standard deviation; Me—median; Min—minimum; Max—maximum; Q1—first quartile; Q3—third quartile; R—range; MT—taxonomic measure; CISS—Coping Inventory for Stressful Situations; NEO-FFI—NEO-Five Factor Inventory.

**Table 3 ijerph-19-09777-t003:** Analysis of the general health of nursing personnel according to the General Health Questionnaire GHQ-30 and subjective emotional control according to CECS.

Variables (*N* = 811)		M ± SD	Me	Min.–Max.	Q1–Q3	R	RQ
CECS	Anger	16.6 ± 4.3	16.0	7.0–28.0	14.0–20.0	21.0	6.0
	Depression	16.4 ± 4.0	16.0	6.0–30.0	14.0–19.0	24.0	5.0
	Anxiety	17.0 ± 4.1	17.0	7.0–28.0	15.0–20.0	21.0	5.0
GHQ-30	Healthy	1.7 ± 1.4	2.0	0.0–4.0	0.0–3.0	4.0	3.0
	Mental health problems	11.8 ± 5.9	10	5.0–30.0	7.0–15.0	25.0	8.0

M—mean; SD—standard deviation; Me—median; Min—minimum; Max—maximum; Q1—first quartile; Q3—third quartile; R—range; RQ—quartile range; CECS—Courtland Emotional Control Care Scale; GHQ-30—General Health Questionnaire.

**Table 4 ijerph-19-09777-t004:** The relationship between the mental state according to the GHQ-30 Questionnaire and the styles of coping with stress according to the CISS Questionnaire.

CISS (Level)		GHQ-30				Chi^2^	*p*
		Healthy (*n* = 437)		Mental Health Problems (*n* = 374)			
		*n*	%	*n*	%		
Task-oriented coping	Low (*n* = 181)	91	20.8	90	24.1	1.4	0.507
	Medium (*n* = 539)	298	68.2	241	54.4		
	High (*n* = 91)	48	11.0	43	11.5		
Emotion-oriented coping	Low (*n* = 46)	40	9.15	6	1.6	75.2	0.000
	Medium (*n* = 612)	357	81.7	255	68.2		
	High (*n* = 153)	40	9.15	113	30.2		
Avoidance-oriented coping	Low (*n* = 57)	39	8.9	18	4.8	12.1	0.002
	Medium (*n* = 633)	347	79.4	286	76.5		
	High (*n* = 121)	51	11.7	70	18.7		
Seeking social contact	Low (*n* = 90)	45	10.3	45	12.0	4.6	0.101
	Medium (*n* = 574)	300	68.65	274	73.3		
	High (*n* = 147)	92	21.05	55	14.7		
Engaging in alternative activities	Low (*n* = 46)	36	8.3	10	2.7	27.3	0.000
	Medium (*n* = 556)	313	71.6	243	65.0		
	High (*n* = 209)	88	20.1	121	32.3		

Chi^2^—Pearson’s chi-square test for independence; *p*—Level of significance, GHQ-30—General Health Questionnaire; CISS—Coping Inventory for Stressful Situations.

**Table 5 ijerph-19-09777-t005:** The relationship between personality traits according to the NEO-FFI Questionnaire and styles of coping with stress according to the CISS Questionnaire (part one).

NEO-FFI		CISS																	
		Task-Oriented Coping						Emotion-Oriented Coping						Avoidance-Oriented Coping					
		Low (*n* = 181)		Medium (*n* = 539)		High (*n* = 91)		Low (*n* = 46)		Medium (*n* = 612)		High (*n* = 153)		Low (*n* = 57)		Medium (*n* = 633)		High (*n* = 121)	
		*n*	%	*n*	%	*n*	%	*n*	%	*n*	%	*n*	%	*n*	%	*n*	%	*n*	%
Neuroticism	Low (*n* = 251)	47	18.7	178	70.9	26	10.4	20	8.20	203	80.86	28	10.94	29	11.55	195	77.69	27	10.76
	Medium (*n* = 373)	88	23.6	245	65.7	40	10.7	14	3.90	285	76.88	74	19.22	16	4.29	293	78.55	64	17.16
	High (*n* = 187)	46	24.6	116	62.0	25	13.4	12	6.67	124	67.18	51	26.15	12	6.42	145	77.54	30	16.04
	Chi^2^	4.2						22.32						14.34					
	*p*	0.38						0.000						0.006					
Extraversion	Low (*n* = 74)	22	29.73	41	55.41	11	14.86	1	1.35	55	74.32	18	24.32	5	6.76	59	79.73	10	13.51
	Medium (*n* = 447)	97	21.70	299	66.89	51	11.41	20	4.47	329	73.60	98	21.92	16	3.58	370	82.77	61	13.65
	High (*n* = 290)	62	21.38	199	68.62	29	10.00	25	8.62	228	78.62	37	12.76	36	12.41	204	70.34	50	17.24
	Chi^2^	3.6						18.49						24.08					
	*p*	0.48						0.001						0.000					
Agreeableness	Low (*n* = 111)	34	30.63	64	57.66	13	11.71	4	3.60	71	63.96	36	32.43	7	6.31	79	71.17	25	22.52
	Medium (*n* = 403)	94	23.33	268	66.50	41	10.17	15	3.72	309	76.67	79	19.60	20	4.96	326	80.89	57	14.14
	High (*n* = 297)	53	17.85	207	69.70	37	12.46	27	9.09	232	78.11	38	12.79	30	10.10	228	76.77	39	13.13
	Chi^2^	9.2						29.31						14.09					
	*p*	0.06						0.000						0.007					
Conscientiousness	Low (*n* = 58)	26	44.83	30	51.72	2	3.45	2	3.45	34	58.62	22	37.93	2	3.45	40	68.97	16	27.59
	Medium (*n* = 376)	96	25.53	248	65.96	32	8.51	12	3.19	286	76.06	78	20.74	20	5.32	303	80.59	53	14.10
	High (*n* = 377)	59	15.65	261	69.23	57	15.12	32	8.49	292	77.45	53	14.06	35	9.28	290	76.92	52	13.79
	Chi^2^	36.9						28.31						11.82					
	*p*	0.000						0.000						0.02					
Openness	Low (*n* = 111)	32	28.83	68	61.26	11	9.91	6	5.26	79	71.93	26	22.81	11	9.91	84	75.68	16	14.41
	Medium (*n* = 457)	106	23.19	310	67.83	41	8.97	26	6.08	332	73.17	99	20.75	32	7	367	80.31	58	12.69
	High (*n* = 243)	43	17.70	161	66.26	39	16.05	14	5.71	201	82.86	28	11.43	14	5.76	202	83.13	27	11.11
	Chi^2^	12.6						11.54						5.18					
	*p*	0.02						0.02						0.27					

Chi^2^—Pearson’s chi-square test for independence; *p*—Level of significance; CISS—Coping Inventory for Stressful Situations; NEO-FFI—NEO-Five Factor Inventory.

**Table 6 ijerph-19-09777-t006:** The relationship between personality traits according to the NEO-FFI Questionnaire and styles of coping with stress according to the CISS Questionnaire (part two).

NEO-FFI		CISS											
		Engaging in Alternative Activities						Seeking Social Contact					
		Low (*n* = 46)		Medium (*n* = 556)		High (*n* = 209)		Low (*n* = 90)		Medium (*n* = 574)		High (*n* = 147)	
		*n*	%	*n*	%	*n*	%	*n*	%	*n*	%	*n*	%
Neuroticism	Low (*n* = 251)	17	6.77	188	74.90	46	18.33	27	10.76	173	68.92	51	20.32
	Medium (*n* = 373)	19	5.09	253	67.83	101	27.08	35	9.38	276	73.99	62	16.62
	High (*n* = 187)	10	5.35	115	61.50	62	33.16	28	14.97	125	66.84	34	18.18
	Chi^2^	12.26						7.2					
	*p*	0.016						0.126					
Extraversion	Low (*n* = 74)	1	1.35	51	68.92	22	29.73	12	16.22	50	67.57	12	16.22
	Medium (*n*= 447)	20	4.47	306	68.46	121	27.07	46	10.29	327	73.15	74	16.55
	High (*n* = 290)	25	8.62	199	68.62	66	22.76	32	11.03	197	67.93	61	21.05
	Chi^2^	11.2						3.97					
	*p*	0.025						0.410					
Agreeableness	Low (*n* = 111)	7	6.31	66	59.46	38	34.23	18	16.22	83	74.77	10	9.01
	Medium (*n* = 403)	16	3.97	281	69.73	106	26.30	39	9.68	298	73.95	66	16.38
	High (*n* = 297)	23	7.74	209	70.37	65	21.89	33	11.11	193	64.98	71	23.91
	Chi^2^	12.78						17.6					
	*p*	0.012						0.001					
Conscientiousness	Low (*n* = 58)	1	1.72	28	48.28	29	50.00	7	12.07	46	79.31	5	8.62
	Medium (*n* = 376)	21	5.59	247	65.69	108	28.72	38	10.11	282	75.00	56	14.89
	High (*n* = 377)	24	6.37	281	74.54	72	19.10	45	11.94	246	65.25	86	22.81
	Chi^2^	27.01						12.34					
	*p*	0.000						0.015					
Openness	Low (*n* = 111)	15	13.51	71	63.96	25	22.52	17	15.32	80	72.07	14	12.61
	Medium (*n* = 457)	20	4.38	313	68.49	124	27.13	52	11.38	324	70.90	81	17.72
	High (*n* = 243)	11	4.53	172	70.78	60	24.69	21	8.64	170	69.96	52	21.40
	Chi^2^	13.08						6.81					
	*p*	0.011						0.146					

Chi^2^—Pearson’s chi-square test for independence; *p*—Level of significance; CISS—Coping Inventory for Stressful Situations; NEO-FFI—NEO-Five Factor Inventory.

**Table 7 ijerph-19-09777-t007:** The relationship between the styles of coping with stress according to the CISS Questionnaire and the level of anxiety, depression and anger suppression according to the CECS Emotional Control Care Scale.

CISS		CECS																	
		Anxiety Suppression Level						Depression Suppression Level						Anger Suppression Level					
		Low		Medium		High		Low		Medium		High		Low		Medium		High	
		*n*	%	*n*	%	*n*	%	*n*	%	*n*	%	*n*	%	*n*	%	*n*	%	*n*	%
Task-oriented coping	Low	40	22.10	120	66.30	21	11.60	43	23.76	115	63.54	23	12.71	55	30.39	109	60.22	17	9.39
	Medium	76	14.10	390	72.36	73	13.54	122	22.63	364	67.53	53	9.83	118	21.89	345	64.01	76	14.10
	High	17	18.68	63	69.23	11	12.09	17	18.68	63	69.23	11	12.09	16	17.58	60	65.93	15	16.48
	Chi^2^	7.23						2.63						8.11					
	*p*	0.124						0.622						0.088					
Emotion-oriented coping	Low	4	8.70	29	63.04	13	28.26	11	23.91	27	58.70	8	17.39	5	10.87	31	67.39	10	21.74
	Medium	108	17.65	437	71.41	67	10.95	141	23.04	412	67.32	59	9.64	152	24.84	388	63.40	72	11.76
	High	21	13.73	107	69.93	25	16.34	30	19.61	103	67.32	20	13.07	32	20.92	95	62.09	26	16.99
	Chi^2^	11.75						4.1						9.15					
	*p*	0.019						0.393						0.058					
Avoidance-oriented coping	Low	9	15.79	42	73.68	6	10.53	16	28.07	36	63.16	5	8.77	13	22.81	35	61.40	9	15.79
	Medium	108	17.06	441	69.67	84	13.27	145	22.91	422	66.67	66	10.43	159	25.12	395	62.40	79	12.48
	High	16	13.22	90	74.38	15	12.40	21	17.36	84	69.42	16	13.22	17	14.05	84	69.42	20	16.53
	Chi^2^	1.75						3.14						7.72					
	*p*	0.781						0.534						0.103					
Engaging in alternative activities	Low	8	17.39	30	65.22	8	17.39	13	28.26	28	60.87	5	10.87	14	30.43	21	45.65	11	23.91
	Medium	93	16.73	391	70.32	72	12.95	124	22.30	373	67.09	59	10.61	131	23.56	360	64.75	65	11.69
	High	32	15.31	152	72.73	25	11.96	45	21.53	141	67.46	23	11.00	44	21.05	133	63.64	32	15.31
	Chi^2^	1.09						1.267						8.603					
	*p*	0.895						0.867						0.072					
Seeking socialcontact	Low	19	21.11	62	68.89	9	10.00	24	26.67	55	61.11	11	12.22	24	26.67	50	55.56	16	17.78
	Medium	93	16.20	402	70.03	79	13.76	131	22.82	382	66.55	61	10.63	135	23.52	268	64.11	71	12.37
	High	21	14.29	109	74.15	17	11.56	27	18.37	105	71.43	15	10.20	30	20.41	96	65.31	21	14.29
	Chi^2^	2.57						2.20						3.26					
	*p*	0.633						0.698						0.515					

Chi^2^—Pearson’s chi-square test for independence; *p*—Level of significance; CISS—Coping Inventory for Stressful Situations; CECS—Courtland Emotional Control Care Scale.

## Data Availability

Data sharing not applicable.

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
