# Peer review of "Analysis of Factors Related to Mental Health, Suppression of Emotions, and Personality Influencing Coping with Stress among Nurses"

_ijerph, 2022, doi:10.3390/ijerph19169777_

Round 1
Reviewer 1 Report
Hi,
The manuscript may be considered for publication after making the suggested corrections. The authors have tested the theory in a timely manner with an appropriate sample. The sample size is more than adequate and authors have presented the constructs with appropriate and up to date references.
You may consider the following feedback.
The current manuscript may be considered for publication after minor changes. The topic is current and has been described in clear way. The description of method, statistical analysis or results have been discussed in an appropriate way. The following points needs to address the review of literature related to problem focused strategies and mental health problems and relationship of personality traits with mental health problems in introduction.
Q:Abstract: Line 26… You may mention that “a demographic questionnaire” was used instead of writing “an own questionnaire”. “An own questionnaire” is not clear and we are not sure what it is measuring.
Q: Abstract: There were no mental health disorders among 53.9% 29 of the respondents. Nurses with mental health disorders according to GHQ-30 were characterized 30 by a high intensity of coping styles focused on emotions (30.2%), avoiding (18.7%) and engaging in 31 alternative activities (32.3%) (p = 0.000). (5)
In the above description in abstract Line: 28, How we categorize population having mental disorder vs. not having mental disorders based on just GHQ-30. The authors did not assess the sample using DSM-5 diagnostic criteria. We may replace mental disorders with mental or psychological problems.
Q:In introduction, the authors did not reviewed literature related to personality traits (e.g., related to introvert, extrovert, conscientiousness, neuroticism, agreeableness). Further, emotion focused coping styles are related to stress but there is no mention of relationship between problem focused strategies with mental health problems.
Q: The authors may consider including more material/ literature in introduction section including paragraphs related to personality traits, stress and mental health problems.
Q: Line 125. Please replace the word materials with “questionnaires”.
Q:Line 125- The following sentence contains the materials two times which needs to be corrected.
The following materials were used to collect 125 the research material:
Q: Line: 405-413: Discussion: The paragraph should be under introduction instead of discussion.
Q:The description of method and analysis section is extensive compared to introduction.
Thank you
Author Response
06.08.2022
Dear Editors,
We take the liberty to thank you and the reviewers for insightful and careful evaluation of our article entitled “Analysis of factors related to mental health, suppression of emotions and personality influencing coping with stress among nurses” by Anna Maria Cybulska, Kamila RachubiÅ„ska*, Marzanna StanisÅ‚awska, Szymon Grochans, Aneta Cymbaluk – PÅ‚oska, Elżbieta Grochans and for allowing us to resubmit a revised manuscript.
The comments helped us to improve the quality of the manuscript. We considered all comments and recommendations and responded to Reviewers’ questions. The correction throughout the manuscript were marked in yellow.
Our responses to the reviews are attached below.
Thank you for your consideration. We look forward to hearing from you.
Sincerely,
Kamila Rachubińska
Response to Reviewers
Open Review
(x) I would not like to sign my review report
( ) I would like to sign my review report
English language and style
( ) Extensive editing of English language and style required
( ) Moderate English changes required
( ) English language and style are fine/minor spell check required
(x) I don't feel qualified to judge about the English language and style
|
|
|
|
Yes |
Can be improved |
Must be improved |
Not applicable |
|
Does the introduction provide sufficient background and include all relevant references? |
(x) |
( ) |
( ) |
( ) |
|
Are all the cited references relevant to the research? |
(x) |
( ) |
( ) |
( ) |
|
Is the research design appropriate? |
(x) |
( ) |
( ) |
( ) |
|
Are the methods adequately described? |
(x) |
( ) |
( ) |
( ) |
|
Are the results clearly presented? |
(x) |
( ) |
( ) |
( ) |
|
Are the conclusions supported by the results? |
(x) |
( ) |
( ) |
( ) |
Comments and Suggestions for Authors
Hi,
The manuscript may be considered for publication after making the suggested corrections. The authors have tested the theory in a timely manner with an appropriate sample. The sample size is more than adequate and authors have presented the constructs with appropriate and up to date references.
You may consider the following feedback.
The current manuscript may be considered for publication after minor changes. The topic is current and has been described in clear way. The description of method, statistical analysis or results have been discussed in an appropriate way. The following points needs to address the review of literature related to problem focused strategies and mental health problems and relationship of personality traits with mental health problems in introduction.
Q:Abstract: Line 26… You may mention that “a demographic questionnaire” was used instead of writing “an own questionnaire”. “An own questionnaire” is not clear and we are not sure what it is measuring.
Q: Abstract: There were no mental health disorders among 53.9% 29 of the respondents. Nurses with mental health disorders according to GHQ-30 were characterized 30 by a high intensity of coping styles focused on emotions (30.2%), avoiding (18.7%) and engaging in 31 alternative activities (32.3%) (p = 0.000). (5)
In the above description in abstract Line: 28, How we categorize population having mental disorder vs. not having mental disorders based on just GHQ-30. The authors did not assess the sample using DSM-5 diagnostic criteria. We may replace mental disorders with mental or psychological problems.
Q:In introduction, the authors did not reviewed literature related to personality traits (e.g., related to introvert, extrovert, conscientiousness, neuroticism, agreeableness). Further, emotion focused coping styles are related to stress but there is no mention of relationship between problem focused strategies with mental health problems.
Q: The authors may consider including more material/ literature in introduction section including paragraphs related to personality traits, stress and mental health problems.
Q: Line 125. Please replace the word materials with “questionnaires”.
Q:Line 125- The following sentence contains the materials two times which needs to be corrected.
The following materials were used to collect 125 the research material:
Q: Line: 405-413: Discussion: The paragraph should be under introduction instead of discussion.
Q:The description of method and analysis section is extensive compared to introduction.
Thank you
RESPONSE: Thank you for this suggestion, we changed it. Your comments helped us to improve the quality of the manuscript. We reviewed and added literature related to personality traits (e.g., related to introvert, extrovert, conscientiousness, neuroticism, agreeableness), stress and mental health problems.
Yours faithfully,
Kamila Rachubińska
Kamila Rachubińska, corresponding author
Department of Nursing, Faculty of Health Sciences, Pomeranian Medical University in Szczecin,
Head: Prof. Elżbieta Grochans
48 Å»oÅ‚nierska St., 71 – 210 Szczecin, Poland
Tel. (091) 48-00-910
Reviewer 2 Report
1. I the opening : explain what you mean by "burnout"
2. Explain the impact of coping strategies and give an example
3. Explain the difference between null hypothesis and alternative hypothesis
318: give an example of "personality traits" and "styles of coping"
412: again explain what you mean by burnout"
421: explain task response
617 explain "emotionally focused style.
In conclusion be clear on how this study could result in better care of nurses and stress prevention within health care
Author Response
06.08.2022
Dear Editors,
We take the liberty to thank you and the reviewers for insightful and careful evaluation of our article entitled “Analysis of factors related to mental health, suppression of emotions and personality influencing coping with stress among nurses” by Anna Maria Cybulska , Kamila RachubiÅ„ska * , Marzanna StanisÅ‚awska , Szymon Grochans , Aneta Cymbaluk – PÅ‚oska , Elżbieta Grochansand for allowing us to resubmit a revised manuscript.
The comments helped us to improve the quality of the manuscript. We considered all comments and recommendations and responded to Reviewers’ questions. The correction throughout the manuscript were marked in yellow.
Our responses to the reviews are attached below.
Thank you for your consideration. We look forward to hearing from you.
Sincerely,
Kamila Rachubińska
Response to Reviewers
Open Review
(x) I would not like to sign my review report
( ) I would like to sign my review report
English language and style
( ) Extensive editing of English language and style required
(x) Moderate English changes required
( ) English language and style are fine/minor spell check required
( ) I don't feel qualified to judge about the English language and style
|
|
|
|
Yes |
Can be improved |
Must be improved |
Not applicable |
|
Does the introduction provide sufficient background and include all relevant references? |
(x) |
( ) |
( ) |
( ) |
|
Are all the cited references relevant to the research? |
(x) |
( ) |
( ) |
( ) |
|
Is the research design appropriate? |
(x) |
( ) |
( ) |
( ) |
|
Are the methods adequately described? |
(x) |
( ) |
( ) |
( ) |
|
Are the results clearly presented? |
( ) |
( ) |
( ) |
( ) |
|
Are the conclusions supported by the results? |
( ) |
(x) |
( ) |
( ) |
Comments and Suggestions for Authors
- I the opening : explain what you mean by "burnout"
RESPONSE: Thank you for this suggestion, we added : Burnout refers to the emotional depletion and loss of motivation that result from prolonged exposure to chronic emotional and interpersonal stressors on the job. Nurse burnout is the state of mental, physical, and emotional exhaustion caused by sustained work-related stressors such as long hours, the pressure of quick decision-making, and the strain of caring for patients who may have poor outcomes. Nursing burnout can lead to feelings of cynicism, hopelessness, and even depression.
- Explain the impact of coping strategies and give an example
RESPONSE: Thank you for this suggestion, we explained the impact of coping strategies and added an example.
- Explain the difference between null hypothesis and alternative hypothesis
RESPONSE: The null and alternative hypotheses are two competing claims that researchers weigh evidence for and against using a statistical test:
- Null hypothesis (H0): There’s no effect in the population.
- Alternative hypothesis (Ha or H1): There’s an effect in the population.
The effect is usually the effect of the independent variable on the dependent variable.
|
BASIS FOR COMPARISON |
NULL HYPOTHESIS |
ALTERNATIVE HYPOTHESIS |
|
Meaning |
A null hypothesis is a statement, in which there is no relationship between two variables. |
An alternative hypothesis is statement in which there is some statistical significance between two measured phenomenon. |
|
Represents |
No observed effect |
Some observed effect |
|
What is it? |
It is what the researcher tries to disprove. |
It is what the researcher tries to prove. |
|
Acceptance |
No changes in opinions or actions |
Changes in opinions or actions |
|
Testing |
Indirect and implicit |
Direct and explicit |
|
Observations |
Result of chance |
Result of real effect |
|
Denoted by |
H-zero |
H-one |
|
Mathematical formulation |
Equal sign |
Unequal sign |
318: give an example of "personality traits" and "styles of coping"
412: again explain what you mean by burnout
421: explain task response
617 explain "emotionally focused style.
RESPONSE: Thank you for this suggestion, we explained everything (burnout, task response, emotionally focused style etc.)
In conclusion be clear on how this study could result in better care of nurses and stress prevention within health care
RESPONSE: Thank you for this suggestion, your comments helped us to improve the quality of the
Yours faithfully,
Kamila Rachubińska
Kamila Rachubińska, corresponding author
Department of Nursing, Faculty of Health Sciences, Pomeranian Medical University in Szczecin,
Head: Prof. Elżbieta Grochans
48 Å»oÅ‚nierska St., 71 – 210 Szczecin, Poland
Tel. (091) 48-00-910
Reviewer 3 Report
Please see attached file.
Conclusions must be expanded, etc.

Author Response
06.08.2022
Dear Editors,
We take the liberty to thank you and the reviewers for insightful and careful evaluation of our article entitled “Analysis of factors related to mental health, suppression of emotions and personality influencing coping with stress among nurses” by Anna Maria Cybulska, Kamila RachubiÅ„ska * , Marzanna StanisÅ‚awska, Szymon Grochans, Aneta Cymbaluk – PÅ‚oska, Elżbieta Grochan sand for allowing us to resubmit a revised manuscript.
The comments helped us to improve the quality of the manuscript. We considered all comments and recommendations and responded to Reviewers’ questions. The correction throughout the manuscript were marked in yellow.
Our responses to the reviews are attached below.
Thank you for your consideration. We look forward to hearing from you.
Sincerely,
Kamila Rachubińska
Open Review
( ) I would not like to sign my review report
(x) I would like to sign my review report
English language and style
( ) Extensive editing of English language and style required
( ) Moderate English changes required
(x) English language and style are fine/minor spell check required
( ) I don't feel qualified to judge about the English language and style
|
|
|
|
Yes |
Can be improved |
Must be improved |
Not applicable |
|
Does the introduction provide sufficient background and include all relevant references? |
(x) |
( ) |
( ) |
( ) |
|
Are all the cited references relevant to the research? |
(x) |
( ) |
( ) |
( ) |
|
Is the research design appropriate? |
( ) |
( ) |
( ) |
( ) |
|
Are the methods adequately described? |
(x) |
( ) |
( ) |
( ) |
|
Are the results clearly presented? |
(x) |
( ) |
( ) |
( ) |
|
Are the conclusions supported by the results? |
(x) |
( ) |
( ) |
( ) |
Comments and Suggestions for Authors
Please see attached file.
Conclusions must be expanded, etc.
RESPONSE: Thank you for this suggestion, your comments helped us to improve the quality of the manuscript.
Yours faithfully,
Kamila Rachubińska
Kamila Rachubińska, corresponding author
Department of Nursing, Faculty of Health Sciences, Pomeranian Medical University in Szczecin,
Head: Prof. Elżbieta Grochans
48 Å»oÅ‚nierska St., 71 – 210 Szczecin, Poland
Tel. (091) 48-00-910